# Analyses of Criminal Judgments about Domestic Child Abuse Cases in Taiwan

**DOI:** 10.3390/children10071237

**Published:** 2023-07-18

**Authors:** Hsiu-Chih Su, Yi-Hxuan Lin

**Affiliations:** Department of Early Childhood Development and Education, Chaoyang University of Technology, Taichung 41349, Taiwan; ivy00112244@gmail.com

**Keywords:** child abuse, domestic violence, criminal judgments, Taiwan

## Abstract

Child abuse has negative impacts on the well-being of children and often leads to adverse consequences, such as suicide, alcohol addiction, depression, and substance abuse. To better understand domestic child abuse in Taiwan, this study analyzed 73 criminal judgments (open-access documents) in which the victims of domestic child abuse were children below the age of 12 from the “Judicial Yuan Law and Regulations Retrieving System” database. There were 73 victims and 91 perpetrators involved. The results indicated that younger children were more likely to be victims of physical abuse, and the majority of death cases were committed by biological parents and cohabiting partners. The perpetrators tended to be young males with lower education. Male cohabiting partners appeared to be a high-risk population for child abuse. Approximately 63% of perpetrators experienced poverty, and 24.7% suffered from marital discord. In the 73 cases, 61.6% of the victims died, 21.9% were mildly injured, and 16.5% were severely injured. The sentencing was related to the level of injury, with perpetrators of mild injury sentenced to less than one year while perpetrators with victim death were sentenced to a longer period of imprisonment. It is suggested that parents at higher risk require greater financial and social support and should be educated on appropriate disciplinary techniques.

## 1. Introduction

Child abuse is a significant global issue, affecting children of all backgrounds and across all countries, irrespective of culture, socioeconomic status, education, income, or ethnicity [1]. According to Article 19 of the UN Convention on the Rights of the Child (1989), States Parties should take necessary measures to safeguard children from physical or mental violence, neglect, exploitation, and abuse, including sexual abuse, through legislative, administrative, social, and educational means. However, a report by the World Health Organization (2019) reveals that one-quarter of all adults have experienced abuse during their childhood [2]. In Taiwan, the Ministry of Health and Welfare estimated that between 2004 and 2018, annually, 4000 to 19,000 children experienced abuse or neglect [3,4].

In Taiwan, the Protection of Children and Youths Welfare and Rights Act (2021) stipulates that no one may abandon, physically or mentally abuse, or subject children and youth to dangerous activities or deceptive behavior that endangers their health. However, despite this, some parents still believe it is acceptable to psychologically and physically punish their children [5]. They hold the deep-rooted belief of “spare the rod, spoil the child,” thinking that discipline for children must involve hitting, scolding, and physical punishment. According to Ministry of Health and Welfare (2022) in Taiwan, there were 149,198 cases of domestic violence reported in Taiwan in 2021, out of which 24,481 involved children and youth (Department of Protective Services,  https://dep.mohw.gov.tw/DOPS/lp-1303-105-xCat-cat01.html, Accessed on 5 July 2023). Perpetrators of child abuse have been found to include parents, cohabiting partners, and relatives who lack parenting knowledge, self-control, and self-esteem, and have a history of childhood abuse or neglect experience [6].

Child abuse is defined as when a parent or caregiver acts or fails to act, causing injury, death, or emotional harm to a child (https://www.childhelp.org/child-abuse/, Accessed on 10 January 2023). There are several forms of child abuse, including neglect, physical abuse, sexual abuse, exploitation, and emotional abuse. Child maltreatment is a significant concern in terms of public health, as it can have long-lasting repercussions for the individuals affected [7]. According to Wang et al. (2020), child abuse has lifelong consequences that negatively impact the physical and mental health of its victims [4,8]. It increases the risk of disorder, depression, smoking, obesity, and substance use disorder [9]. Additionally, children who have experienced abuse are more likely to exhibit higher rates of suicidality, suicide attempts, depression, anxiety, post-traumatic stress disorder (PTSD), alcohol and/or substance use disorders, and episodes of violent behavior [10]. They are also more likely to become abusive parents when they grow up [11]. In Taiwan, Wang et al. (2020) confirmed that child abuse is strongly related to future risk of psychiatric disorders and substance abuse among the general pediatric population [4].

The issue of child abuse is comprehensively addressed by the laws in Taiwan. Despite explicit legal provisions, there remains a significant number of child abuse cases. Therefore, our focus should be on prevention rather than relying solely on punishment as a solution. To gain a better understanding of the factors contributing to child abuse within families, this study analyzes criminal judgments of child abuse cases in Taiwan. By enhancing our understanding of the potential causes and underlying factors, we can significantly improve our ability to prevent child abuse.

The socioecological model, proposed by Urie Bronfenbrenner in 1977, provides a framework for understanding the risk and protective factors of child abuse and neglect [11]. This model includes individual child development, caregivers, family dynamics (microsystem), relations between microsystems (mesosystem), the community and external influences (exosystem), and sociocultural values (macrosystem) [12]. It offers a holistic perspective on child development within an interconnected system [13]. This model recognizes that multiple variables interact within different systems, creating an environment that increases the likelihood of child abuse and neglect [14]. The Centers for Disease Control and Prevention (CDC) uses the model to prevent child abuse and emphasizes the importance of targeting each level for effective and lasting prevention [15].

This study utilizes the socioecological model as a theoretical framework to analyze relevant risk factors. However, due to limitations in the available judicial judgment reports, the focus of this study is primarily on child characteristics and the microsystem of parental and family dynamics. By being alert to signs of child abuse and seeking help, we hope to prevent these tragic incidents from occurring. The objective of this study is to examine the risk factors, circumstances, and consequences associated with child abuse by analyzing judicial cases involving domestic child abuse incidents. Research has indicated that multiple factors contribute to an increased risk of child maltreatment, including child factors, perpetrator factors, and family factors.

Child factors:

A study conducted by Chang et al. (2016) investigated child abuse cases reported to the Child Protection Medical Service Demonstration Center (CPMSDC) in a tertiary medical center in Taiwan between 2014 and 2015 [16]. The study found that the majority of cases involving neglect and physical abuse occurred in children under the age of 6 years. Chung (2021) identified the child’s age as a risk factor, indicating that children aged three years and younger are at a greater risk of child abuse compared to older children [17]. Palusci (2022) also reported that young children and infants are the most susceptible to child physical abuse. According to Palusci (2022), the estimated fatality rate attributed to maltreatment in the United States is 1–2 per 100,000 children annually, with a significant majority (78%) occurring in children under the age of 4, and 44% in infants [1].

Children with disabilities or special healthcare needs are also vulnerable to an increased risk of physical abuse due to the specific behavioral challenges they present. These challenges include noncompliance, aggressive behaviors, communication problems, difficulty meeting parental expectations, and the added stress of managing their health condition. These challenges may not easily respond to traditional methods of correction or discipline, leading caregivers to resort to physical abuse or corporal punishment in an attempt to address these difficulties [13]. Chung (2021) has identified child disabilities as the strongest risk factor associated with a higher likelihood of subsequent allegations of maltreatment [17].

Perpetrator factors:

According to Chang et al. (2016), fathers (50%) and mothers (7%) were the most commonly identified perpetrators of child abuse. Other individuals involved in perpetrating abuse included a parent’s domestic partner, babysitter, and other relatives [16]. However, Giardino et al. (2022) discovered that in cases of physical abuse within the family, both male and female family members accounted for the majority of perpetrators in roughly equal proportions [13].

Several risk factors have been associated with child abuse. These risk factors include intimate partner violence (IPV), depression, poverty, poor housing, and substance abuse. IPV and depression have consistently emerged as two significant risk factors for child maltreatment. Parental depression, in particular, has shown the strongest association with physical abuse, with depressed parents being reported as 3.45 fold more likely to engage in physical abuse compared to non-depressed parents [14].

Kożybska et al. (2022) also identified several factors associated with domestic violence and child abuse [18]. These factors include parental job loss, poverty, involvement in criminal activities, mental illness, substance abuse, household member incarceration, violence against the mother, and separation of parents. It is noteworthy that domestic violence against adults often co-occurs with child abuse. In such cases, the majority of adult respondents who experienced violence were women, while the perpetrators were predominantly men. Risk factors associated with IPV include low levels of education for both men and women, experiences of violence against women during childhood, low socioeconomic status, residing in rural areas, having a higher number of children, and being separated or divorced. Additionally, the partner’s unemployment has also been recognized as a contributing factor. Other studies have supported that criminal history, alcohol and drug use, marital conflict, social isolation, and the presence of the perpetrator’s mental illness are related to both domestic violence and physical violence against children [19].

Poverty is commonly associated with the most prevalent stressors linked to child maltreatment [13]. Caregiver stress and frustration are frequently identified as factors in child maltreatment [20]. Miragoli et al. (2018) found that parenting stress amplifies one’s perception of child behavior and the potential for abuse. Insufficient economic resources, a lack of community and social support, and tense interactions between the child and caregiver(s) can contribute to a higher incidence of maltreatment [21]. Liu and Merritt (2018) found a correlation between high familial financial stress and caregiver aggression [13,22].

Extensive research has supported a strong link between a family’s socioeconomic status and child maltreatment [23]. According to Drake et al. (2022), poverty is associated with both self-reported and officially reported instances of maltreatment. Poverty is strongly associated with child abuse [24]. Age and family structure are related to poverty, with younger people and single-parent families more likely to experience poverty. Poverty also has a negative impact on parenting behaviors [25]. Drake et al. (2022) referenced a study that investigated child maltreatment fatalities by analyzing death certificates. The study connected mortality files of young children from the CDC database with Census information, as conducted by Farrell et al. (2017). The findings revealed that counties with the highest poverty concentration had an abuse fatality rate 3-fold higher than those with the lowest poverty concentrations [26]. Research has shown that poverty is connected to a decrease in positive parenting characteristics, such as hostile parenting [27], harsh discipline [28], and lower parental warmth [29]. A systematic review conducted by Courtin et al. (2019) found that interventions aimed at improving the economic conditions of families with young children lead to a reduction in adverse childhood experiences [24,30]. Yang et al. (2019) established a correlation between the receipt of child care subsidies and a reduced number of maltreatment investigations [31].

Palusci (2022) also mentioned that economic hardship is linked to child abuse. The stress and strain caused by challenging economic conditions can potentially worsen the risk factors for child abuse [1]. Palusci (2022) indicated that increased physical abuse is associated with factors such as poverty, larger family size, unemployment, single parenting, and the presence of an unrelated caregiver. In another study, it was observed that families who experienced fatal child maltreatment were less likely to have received family support, foster care, court-appointed representatives, and case management compared to families whose children did not die from maltreatment [24]. Economic interventions, although not always coordinated with child welfare efforts, have demonstrated preventive benefits.

Kepple et al. (2022) highlighted other risk factors associated with physically abusive parenting practices [32], including alcohol abuse [33], impulsivity, and a low sense of self-control [34], as well as chronic use of cocaine or methamphetamine. Those who reported using two or more substances, especially concurrent marijuana and alcohol use in the past year, were associated with a higher average frequency of physically aggressive parenting behaviors [35]. Substance abuse is undoubtedly a risk factor for child abuse and neglect.

Some research indicates that having a non-biological parent surrogate may be a risk factor for child maltreatment (CM). In a longitudinal study of 644 mother-infant dyads, Radhakrishna et al. (2001) found that homes with a non-biological father figure had a risk of CM that was more than twice as high as families with only the biological mother or both biological parents present, even after considering high-risk variables such as maternal depression, the number of siblings, and maternal education [36]. Research has shown that having a non-biological child from the male partner in the household significantly elevates the risk of femicide (the killing of a woman) in addition to prior instances of domestic violence [19].

Chan et al. (2023) examined risk factors for child abuse and neglect within violent families and found that certain perpetrator-related factors increase the risks of both intimate partner violence (IPV) and child abuse and neglect (CAN). These factors include controlling behavior, accepting violence and mistreatment, substance abuse, involvement in criminal activities, and mental health issues. Moreover, marital conflicts significantly heighten the likelihood of employing harsh discipline in parenting. Family-related factors that contribute to the co-occurrence of IPV and CAN involve poverty, unemployment, stress in parenting, and poor quality of relationships and communication [37].

According to Drake et al. (2022), unemployed parents were found to be more prone to maltreating their children compared to employed parents [24]. Lawson et al. (2020) found that during the pandemic, the relationship between unemployment and child maltreatment continued to exist [38].

Many studies have relied on self-reported questionnaires or interviews to examine child abuse issues, with fewer studies analyzing actual cases of child abuse. Self-report data are subject to limitations such as recall bias, social desirability in parental reporting, and developmental challenges, including the child’s ability to report [24]. Furthermore, it is not feasible to randomly assign children to different experimental groups to investigate associated factors and consequences of child abuse. In this study, the actual contents of judicial reports regarding child abuse within the home were examined.

The research aims to achieve the following objectives:To describe factors related to child abuse in judicial judgments, including child characteristics (age), child risk factors, perpetrator characteristics (identity, gender, age, education, employment, perpetrator risk factors such as negative emotions, lack of parenting knowledge, substance abuse, crime history, depression, mental disability), and family risk factors (poverty, marital discord).To describe the circumstances of child abuse, including the time of occurrence, duration of abuse, number of perpetrators, presence of a third party, type of abuse, hospitalization of victims, and the severity of injury sustained by the victims.To examine the sentences of perpetrators in relation to the severity of child abuse in judicial judgments of child abuse cases.

Definition of Terminology

Child abuse. According to Article 2 of the “Protection of Children and Youths Welfare and Rights Act” (20 January 2020) by the Ministry of Health and Welfare, children are defined as those below 12 years of age, while youth are defined as those between 12 and 18 years old. The Ministry of Health and Welfare has classified child abuse into four categories: abandonment, physical and mental abuse, improper discipline, and witnessing domestic violence. Physical and mental abuse encompasses different types of mistreatment, including physical abuse, emotional abuse, sexual abuse, and neglect. The first three forms of abuse refer to actions that should never occur but unfortunately do. Neglect, on the other hand, refers to a failure to provide adequate care or attention.

According to the website of Childhelp (https://www.childhelp.org/child-abuse/, Accessed on 10 January 2023), child abuse is when a parent or caregiver either acts or fails to act, causing harm or risk of harm to children, including neglect, physical abuse, sexual abuse, exploitation, and emotional abuse. Child maltreatment involves harmful actions or neglect by parents or caregivers that cause serious harm, including death, physical or emotional trauma, sexual abuse, exploitation, or immediate risk of severe harm [17]. It is important to note that this study specifically focused on instances where children under 12 years old were mistreated, injured, or killed by a family member. Sexual abuse cases were excluded due to a lack of detailed information resulting from confidentiality and the need to protect the victims.

Domestic violence. In the Taiwanese context, domestic violence encompasses any form of violence directed towards family members, including children, and is not limited to violence between partners, which differs from the conventional definition of domestic violence. The Domestic Violence Prevention Act in Taiwan (2021) provides its definition of domestic violence as acts of physical, psychological, or economic harassment, control, threat, or other illegal actions against any family member. The Act specifies family members to include spouses or former spouses, individuals in existing or former cohabitation relationships, householders and household members, lineal or collateral relatives by blood or marriage within four degrees of kinship. In this study, we examined cases of domestic violence against children that occurred within the family, resulting in harm or death caused by their parents, cohabiting partners, step-parents, or relatives while under their care.

Criminal judgments. Criminal judgments, as decisions made by a court regarding liability in a legal proceeding, typically provide explanations of specific legal orders. This study only analyzes criminal judgments related to child abuse cases within families, retrieved from the “Judicial Yuan Law and Regulations Retrieving System” database in Taiwan between 1 January 2003 and 31 May 2020.

## 2. Materials and Methods

The documentary analysis method, in which the contents of judicial judgment reports were reviewed and coded, was used in this study. This study first reviewed relevant literature to understand the possible factors influencing the occurrence of child abuse cases within families, serving as the foundation for establishing research hypotheses. Next, using the “Judicial Yuan Legal Information Retrieval System,” we searched for cases of domestic child abuse.

In Taiwan, the “Judicial Yuan Law and Regulations Retrieving System” has provided open access to judicial judgments of criminal and civil cases since 1 January 2003. In order to gain a better understanding of real situations of domestic child abuse, the researchers of this study conducted a search for criminal cases involving victims of child abuse under the age of 12 between 1 January 2003, and 31 May 2020. Sexual abuse cases were excluded from the study due to the confidential nature of judicial judgments and insufficient information. A total of 387 cases were found, and after excluding duplicate cases due to appeals, 73 cases were selected for the study.

The researchers meticulously examined judicial verdicts and created 31 variables that encompassed three categories: related factors, circumstances, and the sentence for cases of domestic child abuse. Firstly, we created variables using the SPSS variable view and recorded the name, data type, values, and value labels. Next, we entered the data into the spreadsheet using the data view. For categorical variables, we utilized value labels to simplify data coding. For instance, we created a variable for child gender with value labels of male and female. The coders carefully read each judicial judgment report and entered the available data for each case. SPSS Statistics software (v.25) was employed for statistical analysis, including descriptive statistics, chi-square, ANOVA, and correlational analyses, to achieve the research objectives.

To ensure inter-rater reliability, two graduate students from the Department of Early Childhood Development and Education, with expertise in child abuse, coded three randomly selected cases for each of the 31 variables. The consistency rate was 97%, with inconsistencies found in three items (education of high school dropout, time of child abuse, and concurrent substance abuse). Subsequently, these variables for the 73 cases were checked, recoded, and modified.

In terms of data analysis, we employed two methods. Firstly, we utilized descriptive statistics to present the occurrences of child abuse and its associated factors. This allowed us to summarize and describe the data in a meaningful way. Secondly, we conducted correlational analysis to examine and illustrate the relationships between the variables. This analysis helped us understand the connections and associations among the different variables under investigation. It is important to note that this study is a correlational study, which means that no causal relationships can be inferred from the findings.

This study adopted a mixed methodological approach, including both quantitative and qualitative analyses. However, due to the qualitative descriptions being solely in Chinese and the inclusion of qualitative data potentially resulting in a significantly longer article, we have chosen to focus solely on quantitative analyses in this paper.

Theoretical Framework:

This study adopted the socioecological model, with a primary emphasis on the characteristics of abused children, perpetrators, and factors within the microsystem of the family. The theoretical framework in Figure 1 comprised of three components. These components encompassed several factors associated with child abuse, including factors related to the child, perpetrator, and family. The second component of the framework described the circumstances surrounding incidents of child abuse, while the third component focused on the legal outcomes of such cases.

## 3. Results

This study analyzed 73 cases of child abuse involving 73 victims and 91 perpetrators.

### 3.1. Factors Related to Domestic Child Abuse

#### 3.1.1. Child Factors

Out of the 73 victims, 74% were under the age of 6, and child deaths accounted for 61.6% of the cases. Younger children were more likely to be physically abused within their families. Of those children under the age of 6, 26% cried constantly and were difficult to soothe, 16.4% were disobedient, and 9.6% had a mental disability. Statistical analysis using chi-square found a significant relationship between age and crying, χ2(6, *n* = 73) = 22.73, *p* = 0.001. Specifically, children who cried constantly and were difficult to soothe were under the age of 6.

#### 3.1.2. Perpetrator Factors

In this study, out of 73 cases, 56 cases (76.7%) involved a single perpetrator, while 16 cases (21.9%) involved two perpetrators, and one case (1.4%) involved three perpetrators, with a total of 91 perpetrators. The perpetrators’ relationship to the victim was varied, with 31.9% being biological mothers, 28.6% being biological fathers, 20.9% being cohabiting partners, 11% being relatives, 4.4% being step-fathers, and 3.3% being step-mothers.

The perpetrator’s identity was significantly associated with the severity of injury inflicted on the child, as determined by a chi-square test (χ2 = 18.795, df = 10, *n* = 91, *p* = 0.043). The majority of cases resulting in death were committed by biological parents and cohabiting partners as indicated in Table 1. Of the 29 mother perpetrators, 14 committed the crime with others. Seven committed the crime with biological fathers, and 7 committed the crime with male cohabiting partners. Furthermore, of the 19 cohabiting partners, 18 were male, indicating that male cohabiting partners may be a high-risk population for child abuse.

Out of 91 perpetrators in 73 cases of child abuse, approximately 57.1% were male while 42.9% were female; 24 had information about their age, with 50.1% being between 20–29, 29.2% being between 30 and 39, 16.5% being between 40 and 49, and 14.2% being above 50 as indicated in Figure 2. Approximately half of the perpetrators involved in cases of domestic child abuse fell between the ages of 20 and 29. The average age of the perpetrators was 30.88 (*SD* = 8.88) years old.

Out of the 36 perpetrators for whom education information was available, 47.2% had graduated from junior high school, 25% had graduated from high school, 16.7% were college graduates, and 11.1% had graduated from elementary school. In general, perpetrators involved in cases of domestic abuse tended to have lower levels of education. The data showed a significant correlation between education level and poverty, with a χ2(3, N = 36) = 10.784, *p* = 0.013, as perpetrators with lower educational levels were more likely to live in poverty. Despite the fact that 93% of the perpetrators were employed, however, because of low education levels, they often experienced poverty problems.

The study identified seven risk factors associated with perpetrators of child abuse in Table 2, which included negative emotions (84.6%), lack of parenting knowledge (68.1%), criminal history such as drug abuse and marital violence (14.3%), substance abuse (11.0%), alcohol addiction (11.0%), depression (15.4%), and mental disability (5.5%). Out of the 91 perpetrators in the study, 49 (53.8%) had two risk factors, 19 (20.9%) had one, 13 (14.3%) had three, 9 (9.9%) had four, and 1 (1.1%) had five risk factors.

Among perpetrators of domestic child abuse, the most frequently observed risk factors were negative emotional responses and a lack of parenting education. Other factors such as depression, a history of crime, substance abuse, alcoholism, and mental disabilities were present but occurred with lower frequency in comparison. Furthermore, it is important to note that a significant majority of perpetrators exhibited multiple risk factors rather than just one.

#### 3.1.3. Family Risk Factors

Out of 73 cases with family risk factors, poverty and marital discord were identified as two main factors (excluding relatives as perpetrators). Of the 73 cases, 43 perpetrators (63%) had poverty issues, and 18 (24.7%) suffered from marital discord.

### 3.2. Circumstances of Child Abuse in Cases of Domestic Violence

#### 3.2.1. Time

According to Figure 3, the majority of incidents of child abuse took place during two time periods. Approximately 35% of the incidents occurred in the afternoon between 12:00 p.m. and 5:59 p.m., while another 35% occurred at night between 6:00 p.m. and 11:59 p.m. The highest percentage of incidents was observed at 11:00 p.m.

#### 3.2.2. Duration

Out of 63 reported cases of child abuse, 72.3% of cases (47 out of 63) occurred within a day, whereas 27.7% of cases (18 out of 63) lasted more than one day.

#### 3.2.3. Number of Defendants

Most cases (76.7%) involved a single perpetrator, while 21.9% involved two perpetrators and 1.4% involved three perpetrators, with a total of 91 perpetrators. The study found that 76.7% of cases involved a single perpetrator, which may be due to the stress of caring for young children alone, leading to child abuse behavior.

#### 3.2.4. Presence of a Third Party

Child abuse often occurred when the perpetrator was alone with the child (79.5% of cases), 20.5% of cases involved the presence of other family members in another room or floor.

#### 3.2.5. Type of Child Abuse

Among 73 cases of child abuse, the methods used included the use of tangible objects (67.1%), bare hands (63%), and parental suicide after killing the child (11%). The number of abuse methods was related to the severity of injury, with the use of both tangible objects and bare hands causing more deaths than just using one method. The cross-tabulation of the types of abuse (physical abuse by hand and using objects) and the severity of harm inflicted on the victims. The use of either type of abuse accounts for 34.2%, while the use of both types accounts for 42.5%. The result of the chi-square test is χ2(4, *n* = 56) = 9.819, *p* = 0.044, which indicates a significant relationship between using both types of abuse and a more severe degree of harm inflicted on the victims.

#### 3.2.6. Hospitalization

Victims were sent to the hospital in 48 cases (65.8% of cases). Approximately 34.2% of perpetrators did not send their victims to the hospital, mainly due to fear or an arrest order for other crimes, such as drug abuse.

#### 3.2.7. Severity of Injury

Out of the 73 cases, 45 victims (61.6%) died, 20 (27.4.9%) were mildly injured, and 8 (11%) were severely injured. The chi-square test analysis results, χ2(2, *n* = 73) = 10.631, *p* = 0.005, indicated that hospitalization was significantly related to the level of injury, with severely injured or deceased children more likely to be hospitalized. However, abused children were often too fragile to survive.

### 3.3. Judicial Sentence of Child Abuse Perpetrators

According to this study, the majority of cases involve the death of the victim, accounting for 62.3%. The second most common type of case involves minor injuries to the victim, accounting for 21.3%. The third type involves severe injuries to the victim, accounting for 16.4%. This indicates that many children lose their lives due to domestic violence. Out of 91 child abuse perpetrators, 86 (94.5%) were given fixed imprisonment sentences, 4 (4.4%) were given life imprisonment, and 1 (1.1%) was given the death penalty for killing both his wife and child. Of those with fixed imprisonment, 31 (36.1%) received less than 1 year, 15 (17.4%) received 1 to 4 years, 23 (26.7%) received 5–9 years, and 17 (19.8%) received 10 years or more. The average sentence length was 71.37 (SD = 8.66) months. A chi-square test, χ2(74, *n* = 151.233), *p* = 0.000, was conducted to examine the relations between the length of imprisonment and the severity of the victim’s injuries, indicating a significant difference. Sentence length was related to the severity of injury, with perpetrators causing mild injury receiving less than 1 year and those causing death receiving longer imprisonment periods. Furthermore, perpetrators who caused death received significantly longer sentences (103.74 months, *SD* = 74.41, *n* = 43) than those who did not (21.64 months, *SD* = 30.78, *n* = 28). In cases of minor injuries to children, most perpetrators were sentenced to less than 1 year, while all perpetrators of severe injuries received less than 5 years. For fatal cases, 40 perpetrators were sentenced to over 5 years in prison.

## 4. Discussion

### 4.1. Factors Related to Domestic Child Abuse

#### 4.1.1. Child Factors

In this study, among the 73 victims of domestic child abuse, 74% were under the age of 6, and child deaths comprised 61.6% of the cases, indicating that younger children were at a higher risk of physical abuse. Young children exhibited traits such as constant crying, being difficult to soothe, disobedience, and mental disabilities. According to Chang et al. (2016), neglect and physical abuse cases predominantly occurred in children under 6 years old [16]. Several researchers, including Chung (2021) and Palusci (2022), identified the child’s age as a risk factor, with younger children being at greater risk of subsequent maltreatment reports [1,17]. Children with disabilities or special healthcare needs were also found to be vulnerable to an increased risk of child abuse due to challenging behaviors such as noncompliance and aggression [13,17]. Chung (2021) specifically highlighted that the presence of child disabilities was the strongest risk factor associated with subsequent maltreatment allegations [17]. The findings of this study partially align with previous research.

Research has found that crying is the most common reason for abuse, as reported in the article (https://newtalk.tw/news/view/2019-01-16/195300, Accessed on 5 July 2023). Difficulty in soothing and calming was found to be more prevalent among children aged 0–6 years old. The triggering factors for child abuse are often associated with common parenting challenges, such as a child’s crying, toileting issues, discipline, and arguments. Therefore, a child’s persistent crying and difficult-to-soothe behavior are important risk factors for child abuse. Thompson (2022) suggested that social support and monitoring of parental conducts to help them cope with stressful situations are important to prevent child abuse [39]. Kimbrough-Melton (2022) also advocated for providing support to higher-risk families, including parenting assistance and instrumental help such as childcare, as a means to address and change child abuse behavior [40].

#### 4.1.2. Perpetrator Characteristics

This study revealed a significant relationship between the identity of the abuser and the severity of the victim’s injuries. In cases where the victim died, the majority of the abuse was committed by biological parents or cohabiting partners. The perpetrators tended to be young males with lower education and inadequate income. Additionally, they often exhibited negative emotions, lacked parenting knowledge, and had a criminal history that included drug abuse, marital violence, substance abuse, alcohol addiction, depression, and mental disability.

In Taiwan, parents sometimes use physical punishment to discipline children, and Article 1085 of the Civil Code allows parents to discipline their children within necessary limits. This blurs the distinction between physical punishment and abuse, which can have lasting traumatic effects on children. When faced with crying or disobedient children, perpetrators who were prone to negative emotional responses and lack of parenting knowledge were more likely to resort to punishment or physical force to control or stop the child from crying. As a result, there was a likelihood of child injury and abuse occurring.

Among the 19 cohabitants, 18 were male, indicating that male cohabiting partners may be a high-risk group for child abuse. Kożybska et al. (2022) found that the perpetrator of domestic violence was usually a man, and violence against adult often co-occurred with children [18].

Among those who provided age information in this study, 50% fell between the ages of 20 and 29, indicating that many children from families within this age group are under six years old. There is a higher risk of child abuse associated with individuals who are young, single, or non-biological parents. (https://www.psychologytoday.com/us/conditions/child-abuse, Accessed on 5 July 2023). The low age range of abusers in this study may be due to two reasons. Firstly, the abused children in the study were under 12 years old, so their caregivers were relatively young. Secondly, sexual abuse cases involving children and adolescents, generally aged 12 to 18 (which were excluded from this study), may involve relatively older caregivers. Drake et al. (2022) found that young people are more likely to experience poverty, which, in turn, has negative impacts on their parenting behaviors. As a result, it is important to provide additional support to younger parents who may face increased financial stress and have less knowledge about effective child rearing [24].

This study found a significant relationship between the educational background of abusers and economic poverty. Abusers with lower levels of education often have lower socioeconomic status and are more likely to face economic difficulties and stress. Despite the majority of perpetrators having employment, their income is typically low or unstable, and they may have financial burdens such as debt at home. In situations where the caregiver’s income is insufficient and the child is crying or in a negative mood, frustration can intensify, leading to the decision to harm the child. Research by Li et al. (2008) supports these findings by demonstrating that caregivers with lower levels of education are more susceptible to experiencing low moods. Consequently, they may resort to punishment and strict measures to halt their children from crying or displaying negative emotions. Additionally, these caregivers may have limited parenting knowledge, unrealistic expectations of their children, and a limited understanding of child development. These factors contribute to difficulties in disciplining and managing their children’s behavior, ultimately increasing the risk of child abuse [41].

Based on the study conducted by Sung (2019), it was found that perpetrators of child abuse often lacked self-control, had inadequate parenting knowledge, were unaware of appropriate ways to discipline or educate their children, and had unrealistic expectations and poor understanding of child development. To prevent child abuse, it is suggested to provide parent education programs or parenting hotlines to enhance parenting knowledge. Moreover, addressing other risk factors such as family poverty, substance abuse, and mental health issues should also be considered to make intervention programs more effective [42].

In cases of child abuse, parents may have other risk factors that prevent them from performing their parental duties, such as poverty, substance abuse, mental illness, domestic violence, and child behavior problems. If structural factors or risk factors are not addressed, the effectiveness of any intervention plan will be greatly reduced [43]. To prevent this, family members should pay close attention to caregivers who are caring for children with such characteristics and avoid leaving them alone with caregivers who have risk factors, especially when under stress or exhaustion.

#### 4.1.3. Family Risk Factors

Out of 73 cases with family risk factors, poverty and marital discord were identified as two main factors (excluding relatives as perpetrators). The study showed that the perpetrators had low education levels, and despite the majority being employed, they often faced poverty problems. Authorities discovered that the majority of families involved in child abuse cases had experienced significant life-altering challenges such as unemployment, child neglect, substance abuse, and marital separation [44]. Sawant and Dsilva (2020) found that children are greatly affected by marital conflicts, which have been found to have a significant impact on their well-being. Studies have indicated that these children often experience a higher incidence of illness. In certain marriages, the inability to effectively resolve conflicts can escalate to instances of abuse and, in some cases, even more severe and devastating consequences [45].

Financial stress, work stress, and insufficient social support often led to negative emotions and corporal punishment when children cried, which could result in child abuse.

### 4.2. Circumstances of Child Abuse in Cases of Domestic Violence

The results of this study showed that child abuse often occurred when the perpetrator was alone with the child (76.7%). Incidents of child abuse most commonly occurred in the afternoon between 12:00 p.m. and 5:59 p.m. and at night between 6:00 p.m. and 11:59 p.m. The highest percentage of incidents was observed at 11:00 p.m. Chang et al. (2016) also found that the majority of cases were reported during daytime hours (8 a.m. to 5 p.m.; *n* = 166, 45.98%), while the fewest cases were reported after midnight (0 a.m. to 8 a.m.; *n* = 46, 12.74%) [16]. It can be inferred that the time period during which most child abuse incidents occurred is when the caregiver is about to rest. When taking care of a child alone, if the child cries and cannot be soothed, the caregiver may become irritable and may resort to physical force (using bare hands or readily available household objects) to harm the child. The more methods of abuse are used, the more likely it is for the child to die.

The majority of child abuse incidents lasted only one day, while some lasted more than one day. It is speculated that child abuse may have occurred multiple times before, but was only discovered when the injuries were too severe to conceal. Abusers may choose to conceal their past offenses in court to reduce their own criminal punishment. Liu (2021) brought attention to the fact that frontline social workers often bear the blame when incidents of child abuse occur. These workers frequently encounter legal limitations that impede the immediate removal of children from dangerous situations. Despite government authorities conducting review meetings to investigate such cases, they often fail to take effective action and address the underlying problems. The inadequacy of social worker manpower further hampers the efficiency of handling cases, resulting in high turnover rates. Effectively safeguarding the safety and well-being of children requires collaborative efforts involving multiple professions and organizations. Relying solely on the social welfare department and social workers is insufficient. It is crucial to establish coordination among governing authorities and implement institutional changes to address systemic barriers in service delivery. Furthermore, providing adequate and tailored support to social workers enables them to work more effectively with children and families, ultimately achieving the goal of risk management and preventing the recurrence of significant child abuse cases [46].

In this study, the majority (61.6%) of abused children died. Even though the majority (65.8%) of abused children were sent to the hospital for emergency treatment after the abuse occurred. There was a significant correlation between the severity of the victim’s injuries and hospitalization. Victims who suffered severe injuries or died were more likely to be sent to the emergency room compared to those who sustained minor injuries. However, younger children are relatively fragile, and after suffering abuse, they often die despite receiving medical treatment. There were cases where the abuser did not send the victim to the hospital after the abuse, which resulted in the victim’s death, because the abuser was afraid of being caught abusing the child or were wanted for other crimes such as using drugs.

### 4.3. Judicial Sentence of Child Abuse Perpetrators

In this study, a total of 91 defendants were involved in the cases, and 86 of them (94.5%) were sentenced to imprisonment. The length of imprisonment ranged from 4 months to 90 months, with an average of 71.37 ± 8.66 months. The length of imprisonment was significantly related to victim death, with more months of imprisonment in cases where the victim died. Among the cases, 36.1% of the defendants were sentenced to less than one year of imprisonment, 26.7% were sentenced to five to nine years and 11 months, and 19.8% were sentenced to ten or more years of imprisonment. Only four defendants were sentenced to life imprisonment, and one was sentenced to death (for killing his wife and child).

In this study, for cases where the victim suffered minor injuries, most of the perpetrators were sentenced to imprisonment for less than one year. For cases where the victim suffered severe injuries, the length of imprisonment for the perpetrators was generally less than five years. In cases where the victim died, a total of 40 abusers were sentenced to imprisonment for more than five years. The length of the sentence was significantly related to the severity of the victim’s injuries.

Many Taiwanese people considered these penalties too lenient which may not deter people from harming children, and in response, the Legislative Yuan in Taiwan amended “The Protection of Children and Youths Welfare and Rights Act” in 2019 to impose fines of 60,000–600,000 NT dollars and allows the public disclosure of the name of anyone who engages in physical or mental abuse against children or youth. The “Criminal Code of the Republic of China” was also amended on 10 May 2019, to allow for perpetrators who cruelly abuse a child or youth under 18 years of age causing death shall be sentenced to life imprisonment or imprisonment for more than 10 years. We compared the maximum months of imprisonment imposed each year for the 73 cases and found that there was a trend of increasing imprisonment terms from year to year. The Taiwan government continues to make efforts to protect children and ensure that perpetrators are held accountable for their actions, and hopes to reduce the number of cases related to domestic violence and child abuse in the future.

## 5. Conclusions

The occurrence of child abuse within families was related to various risk factors, including the age and behavior of the child, the characteristics of the perpetrator such as their identity, age, education level, negative emotions, and lack of parenting knowledge, as well as family factors such as poverty and marital discord. Abuse was most likely to happen between noon and midnight, when the caregiver was alone with the child, and often involved the use of bare hands or everyday objects to cause harm. In cases where the abuse resulted in severe injuries, the child was often too fragile to survive hospitalization. The severity of the sentence for child abuse was correlated with the level of injury, with perpetrators of victims with severe injury and death receiving longer prison terms.

Based on the above research findings, the following recommendations are proposed. The traditional values and approach of “Spare the rod, spoil the child” should be abolished, and parents should be educated on appropriate disciplinary techniques. The government should provide more support for parents through increased child subsidies, affordable quality child care, and parent education programs. As poverty was identified as the top risk factor for child abuse, younger parents often experience economic difficulties in raising children and lack knowledge about child rearing. The government should increase child care and rearing subsidies. In addition, parents should be encouraged to attend parenting education lessons either through online resources or in person to receive subsidies to improve parenting skills and prevent child abuse.

The public should be alert to signs of domestic child abuse. If they suspect child abuse, such as a child crying loudly for a long time, adults yelling loudly, or sounds of objects falling or breaking in the house, they can immediately report it by calling the Women and Children Protection Hotline to prevent further harm or abuse.

In addition, education and protection service personnel should participate in relevant training activities to increase their sensitivity to detecting suspected cases of child abuse in students at an early stage. They should also report any suspected cases of domestic violence to the local authorities within 24 h. Cooperation between education and protection service personnel is essential for early detection and intervention.

Limitations and future research

This study has several limitations. Firstly, it was based on judicial judgment reports of extreme cases of child abuse within Taiwanese households, which may not capture the full extent of child abuse as only a small portion of cases go through the judicial process. Consequently, the incidence of child abuse at home is likely underestimated, and the findings may have limited external validity, making it difficult to generalize them to non-prosecuted cases of child abuse or the average family. Additionally, the results cannot be generalized to other contexts such as child care settings. Lastly, it is important to note that this is a correlational study, and no causal relationships can be inferred.

Based on these limitations, future studies could conduct large-scale retrospective or prospective studies of child abuse victims and caregivers, utilizing surveys or in-depth interviews to provide further insights into the etiology of child abuse. Furthermore, future studies could explore judicial judgment reports of child abuse in different contexts, such as nursery centers, preschools, home-based caregivers, schools, and after-school care. Factors related to child abuse in other ecological systems, such as societal values, law enforcement systems, and social work intervention programs, could also be further investigated.

## Figures and Tables

**Figure 1 children-10-01237-f001:**
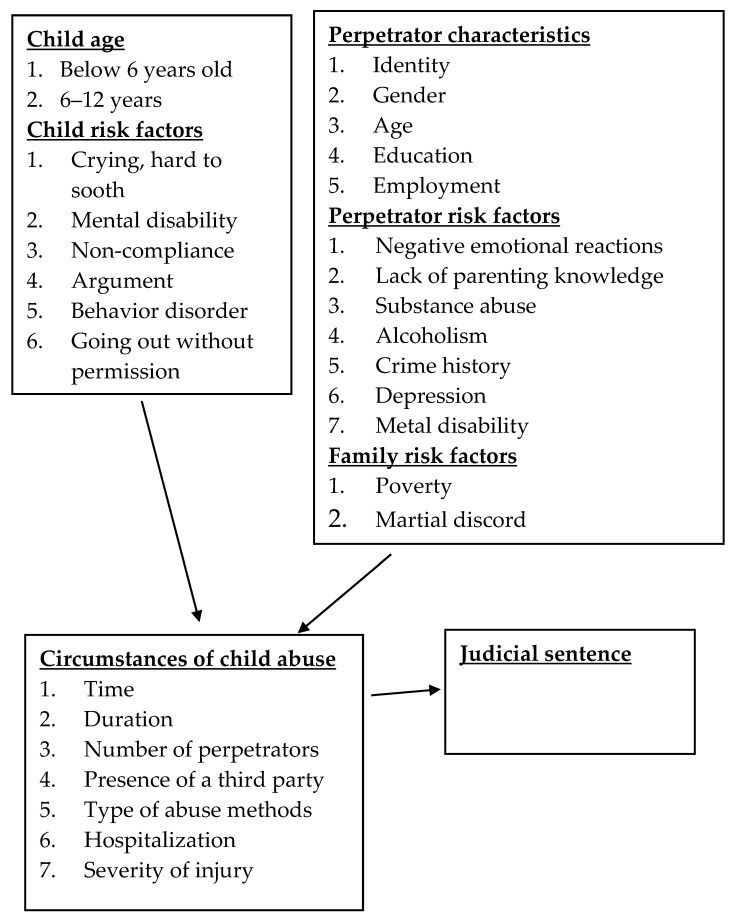
Factors analyzed in the processes of cases of child abuse.

**Figure 2 children-10-01237-f002:**
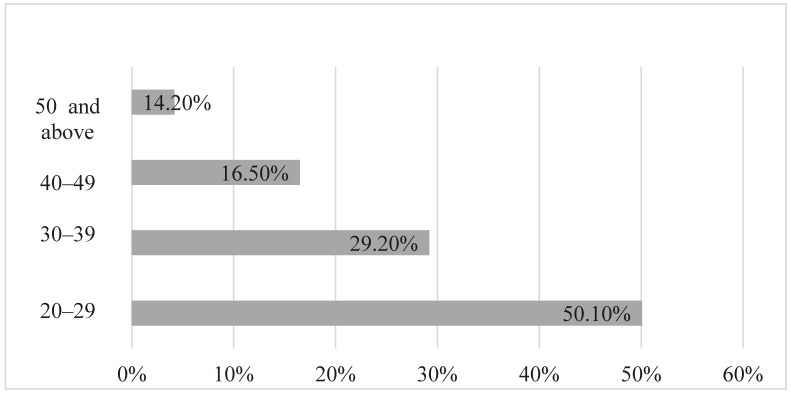
Perpetrator age.

**Figure 3 children-10-01237-f003:**
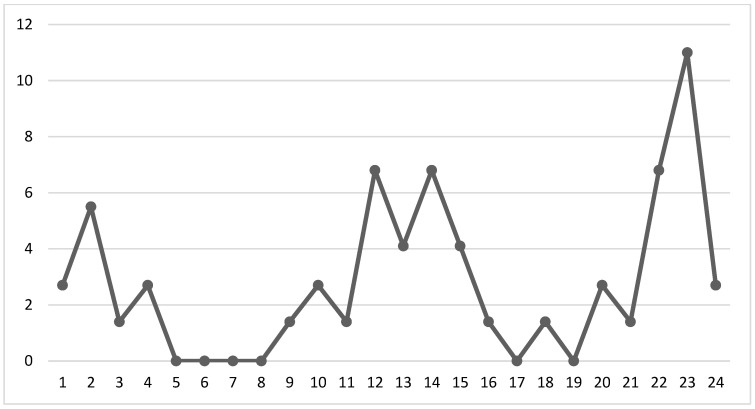
Time of occurrence of child abuse.

**Table 1 children-10-01237-t001:** Perpetrators and Severity of Victim Injury.

	Death	Severely Injuried	Mildly Injuried	*n*/%
Biological mother	20	2	7	29/31.9%
Biological father	17	2	7	26/28.6%
Cohabiting partner	13	4	2	19/20.9%
Relative	6	1	3	10/11%
Step-father	0	3	1	4/4.4%
Step-mother	1	1	1	3/3.3%
Total	57	13	21	91/100%

**Table 2 children-10-01237-t002:** Perpetrator risk factors.

Frequency and Percentage		
Risk Factors	Frequency/N	Percentage
Negative emotional responses	77/91	84.6%
Lack of parenting education	62/91	68.1%
Depression	14/91	15.4%
Crime history	13/91	14.3%
Substance abuse	10/91	11.0%
Alcoholism	10/91	11.0%
Mental disabililty	5/91	5.5%

## Data Availability

Open access judicial judgements are available in “Judicial Yuan Law and Regulations Retrieving System” database at https://judgment.judicial.gov.tw/FJUD/default.aspx, Accessed on 31 May 2020.

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
