# Peer review of "Analyses of Criminal Judgments about Domestic Child Abuse Cases in Taiwan"

_children, 2023, doi:10.3390/children10071237_

Round 1
Reviewer 1 Report
Thank you for the opportunity to review this paper. It offers interesting insights. The paper does need a number of revisions which are attached. I have put comments onto a pdf version of the paper.
Perhaps the most challenging issue is related to terminology - domestic violence v child abuse. It appears that DV is the wrong terminology to be used here as you are really talking about child abuse.
The paper also needs a limitations statement as this type of research is not as generalizable as we might like as this level of CA represents a small portion of those who abuse children. Thus, this is insight into only a limited portion of the perpetrator population. Child welfare authorities tend to see multiples of cases that never make it into then criminal process.
Author Response
Thank you for your kind advice. Please see the attached file for my responses.

Reviewer 2 Report
The article addresses a topic of high social relevance, in a methodological approach that, by using a retrospective documental analysis of court cases, can make better use of the data analysed to better support and discuss the results. Although the issue of domestic violence brings together increased research complexity, there is a panoply of international studies that analyse the risk and protection factors, and the article does not provide a review of the state of the art on the problem of domestic violence involving children. The authors consult current works, but leave unexplored a whole field of literature, namely related to risk assessment of violence against children, including homicide. There are several passages in the introduction where statements are made without substantiating them with several studies. Besides the conceptual definition they present, a theoretical analysis of the main scientific findings on the subject should be carried out, taking as reference inclusion criteria that are close to the study carried out.
The empirical part has a high social and scientific interest, due to the contribution it can give to the prevention of this criminal phenomenon. The method is summarized in practically two paragraphs, at the end of which a paragraph related to the theoretical approach and the variables explored is made again (cf. scheme). There needs to be better organisation of the introductory contents and the method in order to be able to present and then discuss results. The characterisation of the sample is adequately done (although they mention the exclusion of cases of abuse and in lines 287-288 refer to these cases). It is a pity to make use of documental analysis, starting from fundamentally qualitative data and which can support the categorisation of the variables, and reduce, without the reader understanding, the process involved to quantitative data, resulting in a practically described study. It is suggested that the authors elaborate on the process of defining the variables by extracting from the court cases the variables that emerge and not only those that they present but whose criteria they have not explained. A mixed methodological approach is certainly richer. The process from qualitative analysis to quantitative conversion should be carried out by researchers experienced in the subject of violence. The study is of a more descriptive and correlational nature, although many of the conclusions presented do not derive from the analyses made, since they would presuppose another type of analysis. No conclusions can be drawn about the effects of one variable on another, if the relationship between independent variables and dependent ones is not verified. The chi-square test does not establish conclusions about the effects of variables on others (e.g. lines 298-303).
In addition, the authors could improve the presentation of the results. The tables and graphs contribute little to the understanding of the results and could be improved. Other more robust statistical analyses can be performed, building on the results of the association tests performed. The discussion must start from the results and not present conclusions that do not follow from them. At some point, in the middle of the discussion of results, the contributions to the intervention are addressed, when they can be discussed towards the end. At the same time, the conclusions do not assess the limitations of the study carried out. I appreciate the opportunity to review this article, which I believe has the potential to be a good publication if the aspects commented on are improved.
Author Response
Thank you for your insightful comments. I have completed the revisions of the paper. Please find the attached file. If any furthe revisions are needed, please kindly inform me. Thank you once again, and I hope everything goes well with you. Best regards,
I have done

Reviewer 3 Report
Dear Authors,
The appreciation of the article entitled “Analyses of Criminal Judgments About Domestic Child Abuse Cases in Taiwan”:
- The subject of the article is current and relevant and is integrated without Special Issue: “Advanced in Child Abuse and Neglect”. The study provides information on child and perpetrator characteristics, child abuse circumstances and judicial sentences.
However, the article presents a set of weaknesses namely:
- in the abstract the sentence "It is suggested that concepts and practices of "spare the rod, spoil the child" should be banned" should be replaced by explaining the idea using cynical language, for example replacing the expression "spare the rod, spoil the child” by punitive practices;
- the introduction should not be started by a number of an article, so it is recommended to rephrase the sentence;
- in the introduction, a theoretical framework should be elaborated regarding the dimensions analyzed in the empirical study, particularly on the factors associated with child abuse factors related to the child, perpetrator and family;
-after the research objectives there is no sense in a section entitled "Definition of Terminology". The concept of child abuse has been mentioned previously;
- In the empirical part of the article, the authors entitle “theoretical framework” to the factors analysed in the processes of cases of child abuse within families. Thus, a figure 1. Theoretical framework, should be titled by "factors analysed in the processes of cases of child abuse";
- In the discussion of results, new authors (e.g. Chung, 2021 and Kożybska et al., 2022) who are not in the introduction will be introduced. This is due to the lack of theoretical framework of these variables in the introduction.
- The bibliographic reference Kożybska et al. (2022) presented in the results discussion is miscited;
- The references with numbers 12, 13 and 14 are incomplete and the links don't work.
The recommendation is: Reconsider after major revision
Author Response
Thank you for your valuable comments. I have completed the revisions of the paper. Please find the attached file. If any further revisions are needed, please kindly inform me. Thank you once again, and I hope everything goes well for you. Best regards,

Round 2
Reviewer 2 Report
The topic under study, being of high social and scientific relevance, should have an empirical treatment that reflects this relevance. The authors have made several changes, particularly with regard to the identification of risk factors for child development in relation to abuse committed against children in the domestic context, but they still do not reflect the state of the art on the subject, revealing some ignorance about current trends in risk assessment research in this field. This is because they take a traditional risk-centred approach, while risk assessment models incorporate other resilience variables not explained in the article. It is not possible to estimate risk only with a deficit-centred approach. There is mention of studies on impact, several carried out in close geographical areas, but they do not contemplate for example European studies, namely at the origin of the first empirical studies on assessment and risk in this domain, including proposals for methodologies to analyse risk. Therefore, although there is an improvement in the article, there is still no explanation of the state of the art on the subject, which compromises the literature review and, consequently, the discussion of results.
The organisation of the article should be different. How do we start with a theoretical review without first exposing the central concepts that guide the study? How do we enter the method without exposing the research question and having an argument about the gap in knowledge that the presented study responds to? The study now has objectives, but these appear contained between the theoretical review and the concepts, when they should precede the method. Therefore, the article still needs maturation and organisation.
There are writing inaccuracies in the text, for example they state that it is a mixed study, they even mention a qualitative objective in the text ("To examine the sentences of perpetrators in judicial judgements of child abuse cases."), but then this examination is reduced to identifying and classifying the sentences. The wording of the objectives should be more precise. Reducing the study to a quantitative analysis without demonstrating its content makes it less rich when there is material for the opposite. If they are limited to quantification, then the authors should correctly write the objectives as such.
At the level of the method, it is not clarified, not even classifying it as documentary analysis. The data extracted should be rich, but without the reader understanding the coding done, because the criteria for creating the 31 categories applied to 73 cases for analysis are not made explicit. A document analysis process goes through several stages that must be made clear to the reader, it is not enough to say that there was intercoder agreement. The scheme of presentation of the categories, if it considers Bronfenbrenner's theory could perhaps do more justice to this approach, obviously incorporating all the factors that weigh on risk, not just a few. The very single criterion of getting 73 out of 387 for existence of resources should itself be subject to reflection.
The study is more descriptive and correlational in nature, and has improved the descriptive presentation of the results, although the authors could cross-reference the 3 groups of variables in order to deepen the results they have. Limitations and proposals for future studies were added.
I emphasise what was said earlier that given the social and scientific relevance, the article should reflect the same correspondence, so I hope it can be improved.
Author Response
Thank you for your comments. Please see the attached file for the responses.

Reviewer 3 Report
dear Authors,
The authors of the article responded to the requests requested in the review in the various sections of the article. However it suggested:
-In line 649, authors should remove the colon and write directly the proposed recommendations;
- In line 664 the number women and children protection hotline should be removed;
-In line 672, authors should remove the colon.
Congratulations to the authors for the changes made to the article.
Best regards.
Author Response
I have revised the manuscript according to your comments. I appreciate your kindness and constructive feedback. Thank you once again for your support. I am deeply grateful. I wish you all the best. I will upload the final version of manuscript later.
